# Predicting nutritional status for women of childbearing age from their economic, health, and demographic features: A supervised machine learning approach

Md. Mohsan Khudri[1], Kang Keun Rhee[1], Mohammad Shabbir Hasan[2◐], Karar Zunaid Ahsan[3◐]*

1 Department of Economics, Fogelman College of Business and Economics, The University of Memphis, Memphis, Tennessee, United States of America, 2 Department of Computer Science, Virginia Tech, Blacksburg, VA, United States of America, 3 Public Health Leadership Program, Gillings School of Global Public Health, The University of North Carolina at Chapel Hill, Chapel Hill, North Carolina, United States of America

◐ These authors contributed equally to this work.
* zunaid@email.unc.edu

**Data Availability Statement:** This study used data from Demographic and Health Surveys (DHS) for Bangladesh, which are publicly available from the DHS program website (www.dhsprogram.com).

## Abstract

### Background

Malnutrition imposes enormous costs resulting from lost investments in human capital and increased healthcare expenditures. There is a dearth of research focusing on the prediction of women's body mass index (BMI) and malnutrition outcomes (underweight, overweight, and obesity) in developing countries. This paper attempts to fill out this knowledge gap by predicting the BMI and the risks of malnutrition outcomes for Bangladeshi women of child-bearing age from their economic, health, and demographic features.

### Methods

Data from the 2017–18 Bangladesh Demographic and Health Survey and a series of supervised machine learning (SML) techniques are used. Additionally, this study circumvents the imbalanced distribution problem in obesity classification by utilizing an oversampling approach.

### Results

Study findings demonstrate that the support vector machine and k-nearest neighbor are the two best-performing methods in BMI prediction based on the coefficient of determination (R2), root mean square error (RMSE), and mean absolute error (MAE). The combined predictor algorithms consistently yield top specificity, Cohen's kappa, F1-score, and AUC in classifying the malnutrition status, and their performance is robust to alternative standards. The feature importance ranking based on several nonparametric and combined predictors indicates that socioeconomic status, women's age, and breastfeeding status are the most important features in predicting women's nutritional outcomes. Furthermore, the conditional

**Funding:** The author(s) received no specific funding for this work.

**Competing interests:** The authors have declared that no competing interests exist.

**Abbreviations:** ADB, adaptive boosting; AUC, area under the (repository) curve; BDHS, Bangladesh Demographic and Health Survey; BMI, body mass index; CART, classification and regression trees; CT, conditional inference trees; DHS, Demographic and Health Survey; DT, decision tree; EA, enumeration area; KNN, k-nearest neighbors; LLR, LASSO linear regression; LLTR, LASSO logistic regression; LR, linear regression; LTR, logistic regression; MAE, mean absolute error; NB, Naïve Bayes; NN, neural network; OOS, out-of-sample; $R^2$, coefficient of determination; RF, random forest; RMSE, root mean squared error; RT, regression tree; SES, socioeconomic status; SML, supervised machine learning; WENI, Women's Empowerment in Nutrition Index; XGB, eXtreme gradient boosting.

inference trees corroborate that those three features, along with the partner's educational attainment and employment status, significantly predict malnutrition risks.

## Conclusion

To the best of our knowledge, this is the first study that predicts BMI and one of the pioneer studies to classify all three malnutrition outcomes for women of childbearing age in Bangladesh, let alone in any lower-middle income country, using SML techniques. Moreover, in the context of Bangladesh, this paper is the first to identify and rank features that are critical in predicting nutritional outcomes using several feature selection algorithms. The estimators from this study predict the outcomes of interest most accurately and efficiently compared to other existing studies in the relevant literature. Therefore, study findings can aid policymakers in designing policy and programmatic approaches to address the double burden of malnutrition among Bangladeshi women, thereby reducing the country's economic burden.

## Introduction

Malnutrition is a critical global concern that requires a comprehensive and continuous intervention strategy. Almost one-third of the population worldwide experience at least one type of malnutrition: underweight, overweight, or obesity [1, 2]. Nearly 1.9 billion adults globally are either overweight or obese, and about 462 million adults are underweight [2]. By controlling malnutrition, the society can reduce healthcare costs and increase productivity, which would result in saving money and economic prosperity in the long run. Obesity and overweight, resulting from abnormal or excessive fat accumulation, are behavioral and non-communicable risk factors affecting people of all ages, races, ethnicity, and gender. They contribute to critical health conditions, including cardiovascular disease, type 2 diabetes, and cancer which often result in death. In contrast, underweight is associated with lower economic productivity and contributes to higher morbidity and mortality rates [3]. Underweight, overweight, and obesity are major global health problems in developing countries [4]. Many lower-middle-income countries have experienced a high rise in the prevalence of overweight and obesity because of economic growth, industrialization, rapid urbanization, improved transportation system, and a nutrition transition to high-calorie food intake over the last couple of decades.

A study found that improvement in maternal body mass index (BMI) in the past 15 years is accompanied by a diminution in malnutrition status of under-five children [5]. Therefore, this paper focuses on the women of childbearing age as they can play a crucial role in the country's economic development and substantially improve a child's nutritional status. Though the socioeconomic, demographic, and health-protective factors of adult malnutrition are widely known, there is a paucity of information on the prediction of BMI and risks of being underweight, overweight, and obese among women of childbearing age based on those factors. Hence, the likelihood of these risky health behaviors needs to be evaluated and predicted at regular intervals. The study has therefore set three primary objectives. First, this study aims to evaluate the performance of multiple supervised machine learning (SML) algorithms along with the traditional regression algorithms and find the one that best predicts the BMI as well as the nutritional status (underweight, overweight, and obesity) among Bangladeshi women of childbearing age ranging from 18–49 years old. Second, the study ranks key socioeconomic, health, and demographic features related to each outcome based on different SML models.

Thirdly, this study employs conditional inference tree (CT) to predict the outcome based on the interactions among important features and used classification and regression trees (CART), i.e., regression tree (RT) for BMI and decision tree (DT), as a robustness check. The findings will help enhance our understanding of malnutrition, make better predictions of nutritional status, and help policymakers design interventions that will reduce malnutrition at the national and global levels.

There are several economic aspects to justify the significance of this research. Firstly, a child whose mother is obese and has other health complications is likely to suffer from health concerns that may lead to a loss of human capital. Secondly, prenatal and postpartum health care costs for obese mothers and their infants are significantly higher than those with a normal weight [6]. Thirdly, malnutrition significantly contributes to increased absenteeism and unsatisfactory scholastic performance among school-aged children, which is detrimental to the economic growth of a nation [7]. Furthermore, the losses of individuals from insufficient nutrient intake are estimated to be around 10% or more of lifetime earnings in low-income countries [8]. Lastly, job absenteeism complicated by obesity even in developed nations yields an output loss of 4.3 billion dollars each year, and this brings an additional burden to the employers [9, 10]. Lack of skilled human capital is harmful for any developing nation because the cost in terms of productivity loss due to malnutrition can amount to 3 to 16 percent of the GDP [11]. Therefore, a comprehensive investigation to study and predict the risks of being underweight, overweight, and obese in women of childbearing age can potentially inform policy and practice in a developing country like Bangladesh.

## Country background

Bangladesh has been experiencing remarkable progress in economic development for the last three decades. The annual GDP per capita growth reached 5.9% in 2021 compared to 1.11% in 1991, and the population living below the national poverty line has been decreasing [12]. Furthermore, the country has placed fewer restrictions on international trade, reformed the financial sector, and boosted infrastructure investments in geographically delimited economic zones to increase country exports.

The country has also made improvements in socioeconomic and health aspects, such as child and maternal mortality and primary education participation. Child mortality decreased from 146.1 deaths to 29.1 per 1,000 live births between 1990 and 2020, whereas maternal mortality declined from 569 to 176 deaths per 100,000 live births between 1990 and 2015 [13]. In the past decade, enrollment in elementary schools increased from 100% to 120% [12]. Despite all these positive changes, Bangladesh is still struggling to control rising non-communicable diseases due to its fragile health system along with lifestyle and environmental challenges. This alone places a considerable strain on all available resources provided by the government and non-governmental organizations.

In terms of nutrition, Bangladesh is showing a reduction in underweight but an upward trend in overweight and obesity. The cases of underweight in ever-married women have been decreasing over time from 52% in 1996–97 to 30% in 2007 and 12% in 2017–18 [14]. In contrast, the percentage of overweight or obesity in the same cohort has increased to 12% in 2007 and 32% in 2017–18 from 3% in 1996–97 [14]. Past policies on maternal nutrition were skewed towards undernutrition, and this caused a problem to address the rising cases of overweight and obesity. Rural-urban differences in unhealthy BMI categories are also a big concern. The recent 2017–2018 Bangladesh Demographic and Health Survey (BDHS) shows that 13% of rural women are underweight compared to 9% of urban women [14]. On the other hand, 43% of urban women are either overweight or obese whereas 28% of rural women are so. The

recent literature argued that women are more likely to experience overnutrition compared to their male counterparts [15, 16], and this is a serious problem as it is strongly liked to breast and ovarian cancers [17]. Therefore, women of childbearing age (18–49 years) are prone to chronic energy deficiency and malnutrition, and these can bring unwanted birth outcomes. Bangladesh has implemented nutritional programs such as Bangladesh Integrated Nutrition Project and National Nutrition Project to address these concerns. Still, the receptions were mixed and did not show a substantial impact on nutrition outcomes. All these indeed highlight the absence of a strategically built nutrition policy. Our study can shed light on policy managers by providing the necessary information to design a behavioral intervention program.

## Literature review

**Economic and health consequences of malnutrition.** According to the BMI database of the World Health Organization (WHO), on average, adult women are more overweight and obese than their male counterparts. Few other studies also report that women are more likely to be underweight, overweight, and obese than men [18–20]. In addition, married women are more exposed to overweight and obesity than single women [21–23]. Overweight and obesity are linked to a higher risk of gestational diabetes, cesarean birthing, and maternal and early neonatal death during pregnancy [24, 25]. In contrast, the health risks associated with being underweight include weakened immune function, vitamin deficiencies, anemia, increased risk for complications from surgery, fragility, increased perioperative complications, mortality in trauma surgery, etc.

The burdens of being obese or underweight pose critical cost containment conflicts to the health system as well the efficiency for individuals to develop healthier lifestyles [26]. For example, 7% of the total healthcare costs in the European Union come from obesity [27]. BMI is considered one of the attributes in the literature on physical appearance [28]. In some jobs, e.g., sales, physical appearance is likely to be directly related to productivity. Besides, a common perception is obese people are likely to be lazier and less intellectually skilled than their non-obese counterparts [29, 30]. The existing literature provides evidence of potential discrimination made by employers against obese workers [31–33]. It also appears that wages of white females go down due to weight increases [34]. He also observes a negative correlation between weight and wages for other gender-ethnic groups. All these provide evidence of the economic impacts of overweight and obesity. Several studies focus on obesity in developed countries [35–37], however, only a few studies pay attention to this epidemic from developing countries' perspectives [38, 39]. There is a lack of research focusing on the prediction of the nutritional outcomes among women of childbearing age in developing countries. This paper attempts to fill out this knowledge gap by predicting BMI and the risks of being underweight, overweight, and obesity from socioeconomic, demographic, and health features using robust SML models. Our study will aid policymakers in designing the appropriate health policies and strategies, thereby significantly reducing healthcare and economic costs.

**Machine learning in applied economics and public health.** There is an increasing interest among applied economists and public health researchers in using the machine learning (ML) approach to predict human behavior and health outcomes, forecast energy and financial variables, and solve complex real-world problems [40–44]. Although slowly, ML methods are gradually becoming the topic of interest in a rapidly growing methodological literature in economics [45, 46]. The reason is its great potential for extending an econometrician's toolbox, which solves a different problem. ML methods are a subset of artificial intelligence that combine traditional econometric techniques and improve their performance over time, learning the underlying pattern of the data. These techniques are particularly robust in estimating

human behavior and sentiment due to the increased heterogeneity of the underlying data [47]. Although ML does not focus on some formal properties that economists are concerned such as large sample properties of estimators, consistency, normality, efficiency, and identifying causal relationships, its demonstrated ability to outperform conventional methods on specific data sets with respect to out-of-sample (OOS) predictive power is valuable in practice.

A number of economists discuss the benefits of SML techniques for regression, point out the existing problems in economics where ML methods are apposite and emphasize the recent developments in adapting ML for addressing causal questions and policy implications in economics and technical econometrics research [48–51]. Recently, one study adopted eXtreme gradient boosting (XGB), logistic regression (LTR), and random forest (RF) to predict bank failures in the United States [52]. In another study, a variety of SML methods were applied to predict the demand for salty snacks based on sales data, and findings suggested that SML can perform superbly in demand prediction [53]. Besides, there is a debate that ML can be useful in the first-stage estimation of instrumental variables because the ordinary least squares tend to overfit a model [54]. LASSO linear regression (LLR) was used by Wüthrich and Zhu to check whether it could mitigate the omitted variable bias [51]. One study suggested that ML could investigate the treatment effect with respect to observable variables [46]. An economic policy has both causal and prediction aspects. However, there are also occasions in which causal inference is unnecessary. Teacher employment was employed as an example by Mullainathan and Spiess to back this claim [49]. The study indicated that, even if the main goal is to estimate the impact of hiring an additional teacher, the type of teacher a school should pursue is based on prediction, which requires ML.

Another study area is to compare the performance between ML algorithms and regression methods. A study attempting to predict women's height based on the socioeconomic status (SES) of 66 low-and-middle-income countries reported that its best ML algorithm performed slightly better than regression methods and exhibited a lack of nonlinearity between height and SES [55]. Another study employed the regularization feature of LASSO to construct an index called the Abridged Women's Empowerment in Nutrition Index (A-WENI) [56]. They reduced the original 33 categories to 20 and showed that estimates from A-WENI are like those from the original Women's Empowerment in Nutrition Index (WENI) and suggested that the shorter index can help measure nutritional empowerment.

Recent studies in public health prefer ML to traditional methods because of its leverage of data characterization and prediction simultaneously [57]. SML techniques have been extensively used in obesity research because of their capacity to predict with unprecedented precision [58–61]. Using three SML algorithms: support vector machine (SVM), RF, and XGB, a study predicted obesity based on food sales data and documented that baked goods and flour followed by cheese and carbonated drinks are strongly associated with obesity [62]. Using the same three algorithms mentioned above, one study endeavored to predict the mortality from COVID-19 [63]. The study showed that age, oxygen levels, and the type of patient encounter are strongly related to mortality from COVID-19. With a small sample, one study based in Bangladesh employed nine SML algorithms to predict the risks of obesity [64]. Their experimental results show that LTR achieves the highest accuracy out of the nine classifiers. Another study from Bangladesh reported that an RF-based classifier outperformed the other four algorithms [65].

**Differences between econometrics and supervised machine learning.** There are key diofferences between traditional econometrics and SML. First, traditional econometrics focuses on finding a variable that is significant in predicting an outcome. On the other hand, SML algorithms take two steps: finding a model to use and regularizing the model of choice based on cross-validation [49, 55]. One of the strengths of SML is that it predicts the outcome

of interest using a variety of methods. There is no specific rule for following a particular conditional distribution to predict an outcome [46]. Unlike linear regression (LR), some SML techniques do not make parametric assumptions about the variables and allow more flexibility in modeling relationships. Therefore, this will enable researchers the freedom to choose the algorithm of their choice. Therefore, establishing a causal relationship is not the primary goal of supervised learning. Instead, our interest is in finding an algorithm that yields the maximum prediction accuracy and the smallest prediction error.

Second, while traditional econometric techniques can indicate variables associated with a certain health condition, it does not automatically yield a possible interaction between two or more variables related to the disease. In contrast, it is possible to find an interaction between predictors and how they predict the likelihood of exposure to a critical health condition with either decision or RT. Therefore, identifying those variables and their interaction is crucial in building an appropriate intervention strategy.

The third difference lies in the data mechanism. In ML, the data are split into an in-sample portion to train a model and an OOS portion to test its performance to find an algorithm with the smallest prediction error. The performance of the algorithm is evaluated based on the latter because SML algorithms seek to identify functions that can perform well with the OOS dataset. This approach is different from microeconometrics as it does not split the data set into training and test sample for analysis. The focus is more geared toward reducing the overall bias of the model [66].

The fourth difference between these two approaches is data-driven modeling. SML is more concerned about overfitting a model. More features can be inserted into an estimation model to increase the predictive power captured by the $R^2$. However, this is not recommended as it can result in an overfitted model with serious concerns. An overfit model tends to have a low bias, and the in-sample error value could get close to zero. However, the OOS error and variance will be large as the predictions based on OOS will react sensitively to a small change of regressors. In contrast, a model with less complexity is not overfitted and tends to have a large in-sample bias. Nevertheless, the OOS prediction error will not be big due to the small variability of the model. SML algorithms can be tuned by imposing restrictions, known as regularization, to avoid overfitting in the first place. Regularization imposes a penalty on the model when additional variables are introduced to make the prediction efficient. It can take the form of variable selection, tree size, learning rate, number of neighborhoods, and so forth [49]. The amount of regularization a model requires is decided by cross-validation [55]. As mentioned earlier, SML methods split the data into in-sample and OOS portions. However, a single split is not recommended as it cannot generalize how the model performs with the comprehensive data. Therefore, K-fold cross-validation is implemented. The first step is splitting the data set into K equal-sized portions. Then, K-1 random folds of the data will be used as the training (in-sample) set, and the remaining fold becomes the test (OOS) set. These two steps are performed K times, whereas econometricians would use all observations from the data without making such a split.

Finally, unlike econometrics, most tree-based SML methods such as regression and decision trees along with RF provide a unique feature called variable importance. It ranks the variables that are important in predicting the outcome and provides insight into variables that are strongly correlated with the outcome, whereas the goal of traditional econometrics is to establish a causal relationship between the outcome and the regressors based on significance measures.

## Methods

### Data source and sampling design

This study uses data from the most recent wave of 2017–18 BDHS from the Demographic and Health Survey (DHS) website. 2017–18 BDHS is the eighth wave of the survey, conducted

between October 2017 and March 2018. One of the challenges of the DHS data is that it neither contains information on income nor on consumption expenditures of households. We overcome this limitation by employing the data on household assets and housing characteristics [67]. Using these data, an asset index is constructed that proxies for wealth and implicitly long-run economic status. The survey also collected information on the household population's demographic and socioeconomic characteristics, including fertility, mortality, educational attainment, employment status, asset ownership, housing characteristics, maternal and child nutritional status, and access to essential services. The survey contains several data files for different units of analysis. The unit of analysis means who or what is being studied. Depending on the data file, the unit of analysis of BDHS can be households, women, men, or children. This paper uses the data of ever-married childbearing women (*de jure* population) aged from 18 to 49 years old for the empirical analysis. Though the original BDHS collect data on women starting from age 15, we consider from 18 years old as girls transition into adulthood then [68].

The survey adopted a two-stage stratified sampling technique to select households and used the sampling frame from the list of enumeration areas (EAs) of the 2011 Population and Housing Census of the People's Republic of Bangladesh, given by the Bangladesh Bureau of Statistics (BBS). Bangladesh comprises eight administrative divisions, such as Barisal, Chittagong, Dhaka, Khulna, Mymensingh, Rajshahi, Rangpur, and Sylhet. The sampling process involved stratifying the sample into regions and breaking down parts further into urban and rural areas. The EAs, also known as clusters and primary sampling units, for the survey are selected in the first stage employing the probability proportional to size sampling technique. All clusters are approximately of equal size in terms of area. The BDHS 2017–18 includes 672 EAs. A household listing prepared in all selected clusters is utilized as a sampling frame to select households in the second stage. Using an equal probability systematic sampling technique, 30 households, on average, are drawn from each selected cluster [14].

## Variable description

The primary outcome variable of this study is the BMI. The survey computes the BMI using reported data on height and weight of individuals by dividing the weight in kilograms by the square of the height in meters. It is a standardized method for determining whether the body weight and the amount of body fat anyone has been in a healthy range. Following the definition of the WHO [69], we create three categorical outcomes classifying the respondents as being underweight ($BMI < 18.5 \ kg/m^2$), overweight ($25 \leq BMI \leq 29.99 \ kg/m^2$) and obese ($BMI \geq 30 \ kg/m^2$). Overweight and obesity variables are also generated following the Asian BMI cutoffs for robustness analysis. Asians tend to have a higher incidence of cardiovascular diseases and diabetes even at lower BMI levels, and the recommended overweight and obesity cutoffs for Asians by WHO expert consultation are $23–27.49 \ kg/m^2$ and $\geq 27.5 \ kg/m^2$ [70]. The socioeconomic features we consider in this paper include the household SES, educational attainments of women and their partners (years), partners' occupation, and the current employment status of women. The household's SES is measured by the wealth index and classified into three groups: low, middle, and high. The wealth index is a living standard measure constructed using principal component analysis. Selected household assets and characteristics: cars, bicycle, TV, usable land, floor type, source of drinking water, type of fuel, electricity, sanitation facilities, and construction materials are used to create a wealth index [71]. The value of the wealth index relies on the assets mentioned above owned by households and housing characteristics. However, this study does not suggest asset index as a measure of current welfare or poverty in this paper. Instead, we use it as a determinant of malnutrition among women, which depends on the long run as well as current SES. Households are unlikely to respond to

temporary shocks by women's malnutrition. The asset index is considered a proxy for wealth, hence for SES in the long run. The educational attainments take four equivalent categories: received no education, finished primary education, completed secondary education, and higher education degree such as college or graduate degree. The occupation of women's partners is categorized into three groups: unemployed, blue-collar jobs, and white-collar jobs. Those jobs are defined as blue-collar that require physical labor, e.g., agriculture, skilled and unskilled manual work; and white-collar jobs consider all desk jobs, e.g., professional, technical, managerial, and business. The unemployed category includes unemployed, student, and retired women. Lastly, employment status shows whether a woman currently has a job or not.

This paper also controls for marital status, age of women, age during their first childbirth, age when they first lived together with their partners, partner's age, contraceptive method, region, residence, religion, breastfeeding practice, abstaining status, amenorrheic status, access to piped water, open defecation, medical assistance, the total number of children ever born, access to media, family planning service, and the number of household members. A woman is defined as currently single if she is divorced, separated, or widowed during the interview. Women's age is categorized into three groups: 18–25, 26–39, and 40+. The partner's age is divided into three categories: 18–30, 31–49, and 50+. The contraceptive method variable asks the respondents if they use any contraceptive methods or not. An indicator variable is created to point out whether a woman used any of these methods or not. The eight administrative divisions of Bangladesh are classified into three regions based on their geographical locations following [72]: west (Rangpur, Rajshahi, and Khulna), center (Mymensingh, Dhaka, and Barishal), and east (Sylhet and Chittagong). The residence variable looks at whether a woman is from an urban or a rural area. Breastfeeding practice is a dummy variable created by looking into the entries in the maternity history for children born in the last three to five years and if any child is still being breastfed. If no child was born in the previous three to five years, the respondent is assumed not to be breastfeeding. Abstaining status is an indicator variable created from the maternity history by checking if the respondent had been abstaining from sexual relations since the last birth or not. Amenorrheic status indicates if the respondent is currently postpartum amenorrheic. The variable is constructed from the maternity history by checking if the period returned after the last birth. Religion specifies a woman's religious affiliation, and our sample shows most are Muslim. We have three variables for getting medical assistance, and each has a specific question: getting permission to go, receiving financial assistance needed for treatment, and whether the distance to a health facility is manageable or not. The media variable is created by merging three variables: watching TV, listening to Radio, and reading the newspaper. If a woman has access to any of them, we consider that she has access to media. Lastly, family planning is a variable that shows whether a household heard any information about family planning through media.

## Data pre-processing and SML algorithms

Variables of interest are extracted from the BDHS 2017–18 wave. In the pre-processing stage, 5,365 women are removed out of 20,127 from the analysis. These women met at least one of the following criteria: (1) women aged 15–17 ($n = 651$) (2) pregnant during the survey ($n = 1,013$); (3) missing information on BMI and other covariates ($n = 3624$) and (4) irrelevant observations and outliers ($n = 77$). Before prediction, the relationships between categorical features and nutritional outcomes are evaluated using the Pearson $\chi^2$ statistic corrected for the complex survey design with the second-order correction of Rao and Scott [73, 74].

For the continuous outcome, the following methods are applied: LR, LLR, RT, RF, CT, XGB, k-nearest neighbors (KNN), nonlinear SVM, and neural network (NN). One of our

categorial outcomes, obesity, shows an imbalanced distribution of its underlying classes. A major problem of an imbalanced category is that it could yield either inaccurate or biased estimates of the measures such as accuracy and precision. This issue is circumvented by applying oversampling technique suggested by Lunardon and colleagues [75]. For the categorical outcomes, the following SML algorithms are implemented: LTR, LASSO logistic regression (LLTR), DT, RF, CT, adaptive boosting (ADB), XGB, KNN, nonlinear SVM, NN, and Naïve Bayes (NB).

The approaches under SML fall into four groups of predictors: parametric, nonparametric, mixed, and combined. LR, LLR, LTR, and LLTR estimate coefficients using parametric functions to predict an outcome. Nonparametric approaches include KNN and tree-based methods such as RT and DT that predict outcomes from a particular threshold of a variable. NB and SVM can be considered either parametric or nonparametric. The former can be regarded as parametric as it requires covariates to be conditionally independent and normally distributed. But it can be nonparametric, taking into account that a probability distribution does not need to be specified [76, 77]. SVM is generally treated as a parametric approach as the hyperplane has a functional form to separate the classes. However, it can be nonparametric depending on the kernel choice [78]. NN is a mixed predictor algorithm as each regressor is connected to a hidden layer to predict an outcome. Lastly, RF and ADB are combined predictors since they are weighted combinations of tree-based predictors. A brief description of all the algorithms used in this study is provided in Supplement A in S1 File.

Throughout the analysis, 80% of the data are used as the IS data and the remaining 20% as the OOS. Nevertheless, splitting the data into two parts may result in random sampling errors [49]. To optimize OOS prediction, both 5-fold and 10-fold cross-validations are used that allow us to find the optimal parameters of each model based on in-sample data and test the performance subsequently on the OOS data. The methodology of this paper is delineated in Fig 1.

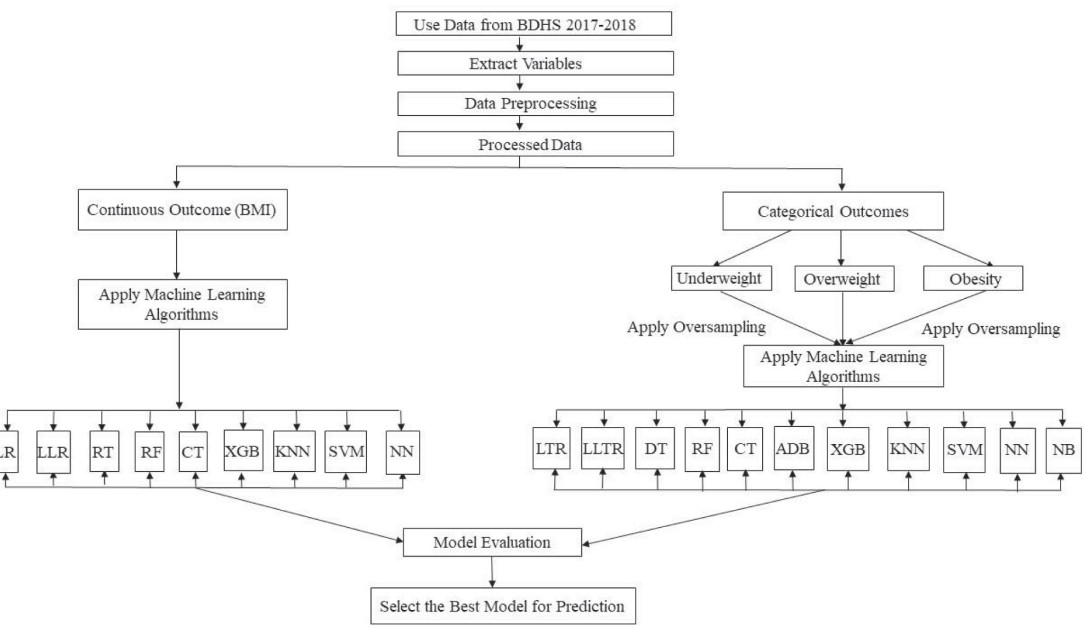

**Fig 1. The methodological flowchart of the study.**

## Evaluation metrics

Since this study consists of both regression and classification tools, a variety of evaluation measures are employed. The root mean squared error (RMSE), the mean absolute error (MAE), and the $R^2$ values are used to evaluate the model performance for regression models. Both RMSE and MAE can be considered standard errors of the prediction. It is preferable to have a lower value for those measures as they indicate that regression predictions are close to the actual outcome values. $R^2$ is the variability of the outcome explained by changes in regressors and therefore captures the predictive power of a model. All these measures are applied to the OOS data and determine the algorithm that performs the best.

For the classification setting, traditional measures such as specificity, Cohen's kappa, area under the repository curve (AUC), and F1-score are used in the ML literature. Same as the evaluation metrics, all these classification measures are analyzed based on the OOS data. Specificity indicates the percentage of true negatives over the sum of true negatives and false positives. A higher specificity value implies a lower number of false positives in the classification. Cohen's kappa is the reliability measure of a classification model, and it is desirable to have it between 0.21 and 1 [79]. A Cohen's kappa value of 0.21 indicates that a classification model is fairly classifying the instances. A value of 1 suggests a perfect classification. The area under the repository curve (AUC) shows the trade-off between the false positive and true positive rates at different threshold values. It is desirable to have an AUC value greater than 0.5 because an AUC of 0.5 implies that a classification model is no better than a fair random guessing of two classes. Lastly, F1-score is a weighted average of precision and recall. It is useful when there is an unbalanced distribution of the cases. The details on all evaluation measures of our study are provided in the Supplement A in S1 File.

## Ethics approval

Demographic and Health Survey Data collection procedures for the 2017–18 BDHS were approved by the Institutional Review Boards of the ICF International, Rockville, MD, USA, and Bangladesh Medical Research Council, Dhaka, Bangladesh. Informed consent was obtained from all respondents in the survey before asking questions, and separately before obtaining biomarker and anthropometric measurements. Respondents who did not provide consent were excluded from the analysis for the current study.

## Results

### Descriptive statistics of study participants

Table 1 shows the descriptive statistics (mean or percentage) and bivariate analysis results of women's BMI, underweight, overweight, and obesity status by their socioeconomic, health, and demographic characteristics from BDHS 2017–18. Estimates are reported incorporating sampling weights from the survey and adjusting for strata and clusters in the sample. The average BMI of women with higher education, higher socioeconomic levels, white-collar jobs, highly educated partners, piped water, or living in urban areas is higher than that of women with other characteristics. On average, the sampled women have a BMI of 23.5 kg/m$^2$, ranging between 22.0 and 25.4 kg/m$^2$. Bivariate analyses show a significant difference (p-value < 0.001) in average BMI across all features except contraception.

Overall, about 27% of the sampled women appear to be overweight. This class is significantly (p<0.05) associated with all features except for current contraception use, permission to go to medical, exposure to family planning messages, and religion. Striking disparities are observed in the prevalence of underweight across different subgroups, such as women with

**Table 1. Bivariate analysis between ever-married women's nutritional outcomes, socioeconomic, health, and demographic characteristics from BDHS 2017–18.**

| Features | Nutritional Status | | | | | | | |
| --- | --- | --- | --- | --- | --- | --- | --- | --- |
| | BMI (kg/m$^2$) | | Underweight | | Overweight | | Obesity | |
| | Mean | p-value | % | p-value[3] | % | p-value[3] | % | p-value[3] |
| Overall | 23.46 | | 10.97% | | 26.64% | | 6.81% | |
| **Education** | | <0.001[1] | | <0.001 | | <0.001 | | <0.001 |
| No education | 22.60 | | 14% | | 22% | | 4% | |
| Primary | 23.19 | | 13% | | 25% | | 6% | |
| Secondary | 23.76 | | 9% | | 28% | | 7% | |
| Higher | 24.65 | | 7% | | 34% | | 11% | |
| **Employment status** | | <0.001[2] | | <0.001 | | <0.001 | | <0.001 |
| Unemployed | 23.96 | | 10% | | 30% | | 9% | |
| Employed | 22.98 | | 12% | | 23% | | 5% | |
| **Socioeconomic Status (SES)** | | <0.001[1] | | <0.001 | | <0.001 | | <0.001 |
| Low | 22.06 | | 17% | | 18% | | 3% | |
| Middle class | 23.34 | | 10% | | 27% | | 5% | |
| High | 24.96 | | 5% | | 36% | | 12% | |
| **Age** | | <0.001[1] | | <0.001 | | <0.001 | | <0.001 |
| 18–25 | 22.18 | | 16% | | 17% | | 3% | |
| 26–39 | 23.86 | | 9% | | 30% | | 8% | |
| 40+ | 23.90 | | 10% | | 30% | | 9% | |
| **Age at first birth** | | <0.001[2] | | <0.001 | | <0.001 | | <0.001 |
| < 20 | 23.29 | | 12% | | 25% | | 6% | |
| 20+ | 23.96 | | 9% | | 30% | | 8% | |
| **Age at first cohabitation** | | <0.001[2] | | <0.001 | | <0.001 | | <0.001 |
| < 20 | 23.39 | | 11% | | 26% | | 7% | |
| 20+ | 24.27 | | 8% | | 31% | | 9% | |
| **Total children born** | | <0.001[2] | | <0.001 | | <0.001 | | 0.004 |
| < 5 | 23.55 | | 10% | | 27% | | 7% | |
| 5+ | 22.77 | | 15% | | 22% | | 5% | |
| **Breastfeeding practice** | | <0.001[2] | | <0.001 | | <0.001 | | <0.001 |
| No | 23.85 | | 9% | | 30% | | 8% | |
| Yes | 22.21 | | 16% | | 16% | | 4% | |
| **Contraceptive method** | | 0.61[2] | | 0.028 | | 0.192 | | 0.299 |
| No | 23.49 | | 12% | | 27% | | 7% | |
| Yes | 23.45 | | 11% | | 26% | | 7% | |
| **Abstaining status** | | <0.001[2] | | 0.664 | | <0.001 | | 0.167 |
| No | 23.49 | | 11% | | 27% | | 7% | |
| Yes | 22.81 | | 10% | | 17% | | 5% | |
| **Amenorrheic status** | | <0.001[2] | | 0.023 | | <0.001 | | 0.162 |
| No | 23.52 | | 11% | | 27% | | 7% | |
| Yes | 22.44 | | 14% | | 15% | | 5% | |
| **Immunization roster entry** | | <0.001[1] | | <0.001 | | <0.001 | | <0.001 |
| 0 | 23.87 | | 9% | | 30% | | 8% | |
| 1 | 22.46 | | 15% | | 19% | | 5% | |
| 2+ | 21.96 | | 12% | | 14% | | 2% | |
| **Height/weight roster entry** | | <0.001[1] | | <0.001 | | <0.001 | | <0.001 |
| 0 | 23.94 | | 9% | | 30% | | 8% | |
| 1 | 22.95 | | 13% | | 23% | | 6% | |

(*Continued*)

**Table 1.** (*Continued*)

| Features | Nutritional Status | | | | | | | |
|---|---|---|---|---|---|---|---|---|
| | BMI (kg/m²) | | Underweight | | Overweight | | Obesity | |
| | Mean | *p*-value | % | *p*-value[3] | % | *p*-value[3] | % | *p*-value[3] |
| 2+ | 22.10 | | 15% | | 15% | | 4% | |
| **Permission to go to medical** | | <0.001[2] | | 0.035 | | 0.268 | | 0.017 |
| No | 23.52 | | 11% | | 27% | | 7% | |
| Yes | 22.98 | | 13% | | 25% | | 5% | |
| **Financial assistance** | | <0.001[2] | | <0.001 | | <0.001 | | <0.001 |
| No | 23.98 | | 9% | | 29% | | 8% | |
| Yes | 22.84 | | 14% | | 23% | | 5% | |
| **Distance to health facility** | | <0.001[2] | | <0.001 | | <0.001 | | <0.001 |
| Not far | 23.77 | | 10% | | 29% | | 8% | |
| Far | 23.05 | | 13% | | 24% | | 6% | |
| **Family planning methods services** | | <0.001[2] | | 0.298 | | 0.668 | | 0.14 |
| Not received | 23.48 | | 11% | | 27% | | 7% | |
| Received | 23.04 | | 12% | | 26% | | 5% | |
| **Partner's age** | | <0.001[1] | | <0.001 | | <0.001 | | <0.001 |
| 18–30 | 22.05 | | 17% | | 16% | | 3% | |
| 31–49 | 23.80 | | 9% | | 29% | | 8% | |
| 50+ | 23.83 | | 11% | | 29% | | 8% | |
| **Partner's education** | | <0.001[1] | | <0.001 | | <0.001 | | <0.001 |
| No education | 22.61 | | 14% | | 21% | | 5% | |
| Primary | 23.05 | | 13% | | 24% | | 6% | |
| Secondary | 23.82 | | 9% | | 30% | | 7% | |
| Higher | 25.06 | | 6% | | 36% | | 12% | |
| **Partner's occupation** | | <0.001[1] | | <0.001 | | <0.001 | | <0.001 |
| Blue-collar job | 23.06 | | 12% | | 24% | | 5% | |
| White-collar job | 24.58 | | 7% | | 33% | | 11% | |
| Unemployed | 23.84 | | 13% | | 33% | | 8% | |
| **Media access** | | <0.001[2] | | <0.001 | | <0.001 | | <0.001 |
| No | 22.34 | | 15% | | 21% | | 3% | |
| Yes | 24.07 | | 9% | | 30% | | 9% | |
| **Piped water** | | <0.001[2] | | <0.001 | | <0.001 | | <0.001 |
| No | 23.35 | | 11% | | 26% | | 6% | |
| Yes | 25.40 | | 5% | | 36% | | 15% | |
| **Open defecation** | | <0.001[2] | | <0.001 | | 0.007 | | 0.046 |
| No | 23.52 | | 11% | | 27% | | 7% | |
| Yes | 22.83 | | 14% | | 23% | | 5% | |
| **Religion** | | 0.018[2] | | 0.088 | | 0.263 | | 0.182 |
| Non-Muslim | 23.09 | | 13% | | 25% | | 6% | |
| Muslim | 23.50 | | 11% | | 27% | | 7% | |
| **Number of family members** | | 0.005[2] | | 0.527 | | 0.006 | | 0.278 |
| 1–5 | 23.54 | | 11% | | 27% | | 7% | |
| 6+ | 23.31 | | 11% | | 25% | | 6% | |
| **Residence** | | <0.001[2] | | <0.001 | | <0.001 | | <0.001 |
| Urban | 24.64 | | 8% | | 34% | | 11% | |
| Rural | 23.05 | | 12% | | 24% | | 5% | |
| **Region** | | <0.001[1] | | 0.203 | | 0.014 | | 0.036 |

(*Continued*)

**Table 1.** (Continued)

| Features | BMI (kg/m²) | | Underweight | | Overweight | | Obesity | |
|---|---|---|---|---|---|---|---|---|
| | **Nutritional Status** | | | | | | | |
| | **Mean** | **p-value** | **%** | **p-value³** | **%** | **p-value³** | **%** | **p-value³** |
| West | 23.21 | | 12% | | 25% | | 6% | |
| Center | 23.58 | | 11% | | 28% | | 7% | |
| East | 23.68 | | 10% | | 28% | | 8% | |

Notes: All statistics are population-weighted using the sampling weight of the BDHS survey

[1] p-values are reported from one-way ANOVA

[2] p-values are reported from the independent samples t-test. For all the categorical outcomes

[3] p-values are reported from the Chi-square test. For all categorical outcomes, the Pearson $\chi^2$ statistic is corrected for the survey design using the second-order correction of Rao and Scott [86] and is converted into an F statistic. The p-value of the corresponding F statistic is reported.

higher education (34%) and no education (22%), those from high SES (36%) and low SES (18%), and breastfeeding (16%) versus non-breastfeeding women (30%). Most of the features considered in this study have a statistical association with obesity. Nevertheless, obesity does not appear to be associated with contraception, family planning, or religion, as is the case in overweight. In addition, obesity is not significantly associated with the number of family members, abstinence, and amenorrhea. Obesity is more prevalent among women with higher education (11%) and those from wealthy families (12%). Women from low socioeconomic backgrounds and those without any education have a lower prevalence (4% and 3%, respectively). A higher percentage of urban women are overweight (34%) and obese (11%) compared to their rural counterparts (24% and 5%, respectively). Conversely, rural women tend to be more underweight than their urban counterparts (12% versus 8%).

## Machine learning algorithm specifications

Table 2 summarizes the ML algorithms used in this study. This study determines the optimal parameter for the algorithms using 5-fold and 10-fold cross-validations to avoid overfitting.

The following sections emphasize performance evaluation measures obtained from the OOS 10-fold cross-validation since the 5-fold results are almost identical. All 5-fold cross-validation results are reported in Supplement B in S1 File (see SB-1 to SB-6 Tables in S1 File).

## Evaluation measures

**BMI prediction.** Table 3 presents the BMI prediction results from all the regression methods investigated in this study. This study uses $R^2$, RMSE, and MAE, which are widely used metrics to evaluate prediction error rates and model performance of regression methods. Table 3 suggests that SVM and KNN are the best-performing models for BMI prediction as they rank first and second in the aforementioned measures. Specifically, the $R^2$ values of SVM and KNN exceed that of LR by 8.6% and 5.5%, respectively. These findings justify the use of SML in this study to predict women's BMI.

**Classification of nutritional status.** Fig 2 compares the predictive performance of classification algorithms for nutritional statuses based on the following four metrics: specificity, Cohen's kappa, AUC, and F1-score. The combined predictor methods, namely ADB, XGB, and RF, show the best performance across all three nutritional status categories by yielding the highest values for all performance metrics considered in this study. Among these three, ADB produces the highest values in the following performance categories: F1-score, Cohen's kappa,

**Table 2. Specifications of SML algorithms.**

| Algorithm | Specifications |
|---|---|
| *Global/parametric predictors* | |
| Linear regression (LR) | Ordinary least squares to fit a linear line |
| LASSO linear regression (LLR) | Type of regularization: *L1*<br>5-fold $\lambda$ = 0.032<br>10-fold $\lambda$ = 0.043 |
| Logistic regression (LTR) | Extension of LR to estimate probabilities<br>Maximum number of iterations = 4 |
| LASSO logistic regression (LLTR) | Type of regularization: *L1*<br>5-fold $\lambda$ = 0.006<br>10-fold $\lambda$ = 0.008 |
| Naïve Bayes (NB) | Probability distribution: Gaussian |
| *Local/nonparametric predictors* | |
| Regression tree (RT) | 5-fold $\alpha$ = 0.007<br>10-fold $\alpha$ = 0.002<br>Number of terminal nodes: 8<br>5-fold and 10-fold depth: 3 |
| Decision tree (DT) | 5-fold $\alpha$ = 0.006<br>10-fold $\alpha$ = 0.002<br>Number of terminal nodes: 7<br>5-fold and 10-fold depth: 3<br>Information measure: Gini index |
| Conditional tree (CT) | Number of terminal nodes: 8<br>5-fold and 10-fold depth: 3<br>Minimum *p*-value: 0.01<br>Information measure: Gini index |
| *k*-nearest neighbor (KNN) | Neighbors for classification: 7 (5-fold), 9 (10-fold)<br>Neighbors for regression: 13 (5-fold), 23 (10-fold)<br>Distance measure: Euclidean |
| Support vector machine (SVM) | Kernel: Radial<br>5-fold and 10-fold *C*: 1<br>Regression support vectors: 10,388<br>Classification support vectors: 8,889 |
| *Combined predictors* | |
| Adaptive boosting (ADB) | Number of iterations: 100<br>5-fold and 10-fold learning rate: 0.14 |
| Random forest (RF) | Number of bootstrapped trees: 100 |
| eXtreme Gradient Boosting (XGB) | Number of iterations: 250<br>Regression depth: 2 (5-fold), 3 (10-fold)<br>Regression learning rate: 0.1 (5-fold), 0.05 (10-fold)<br>Classification depth: 5 (5-and 10-fold)<br>Classification learning rate: 0.15 (5-and 10-fold) |
| *Mixed predictors* | |
| Neural network (NN) | Regression hidden layer number(s): 1<br>Regression network structure: 27-100-1<br>Regression activation function: Rectified linear unit<br>Classification hidden layer number(s): 1<br>Classification network ***structure***: 27-9-1<br>Classification activation function: Sigmoid |

Notes: The *L1* regularization forces some coefficient values to be 0. $\lambda$ is the regularization parameter for LLR. $\alpha$ is the regularization parameter for RT and DT. *C* is the cost of misclassification for SVM.

and specificity. RF achieved the highest AUC across all three classes while achieving the second-highest score across the remaining three metrics. XGB ranks third by producing the second-highest AUC as well as the third-best scores for the remaining three metrics. In

**Table 3. Performance of different regression algorithms in BMI prediction using 10-fold cross-validation.**

| Regression Methods | $R^2$ | RMSE | MAE |
|:---:|:---:|:---:|:---:|
| LR | 22.0% | 3.404 | 2.774 |
| LLR | 21.6% | 3.415 | 2.786 |
| RT | 17.7% | 3.497 | 2.849 |
| RF | 21.9% | 3.439 | 2.807 |
| CT | 14.9% | 3.558 | 2.890 |
| XGB | 22.1% | 3.402 | 2.769 |
| KNN | 23.2% | 3.387 | 2.756 |
| SVM | **23.9%** | **3.372** | **2.718** |
| NN | 14.5% | 3.498 | 2.822 |

Notes: The best performance for each category is in bold. OOS = out-of-sample (test data). Here, LR = Linear regression; LLR = LASSO linear regression; RT = Regression tree; CT = Conditional inference tree; RF = Random forest; XGB = eXtreme Gradient Boosting; KNN = $k$- nearest neighbor; SVM = Support vector machine; NN = Neural network. $R^2$ = R-squared score for goodness of fit, RMSE = Root mean squared error, MAE = Mean absolute error.

underweight classification, the F1-scores of combined predictors outperform LTR by 42.3%, 33.3%, and 17.4%, respectively. Besides those, CT outperforms other methods in most instances except for producing slightly lower specificity values in overweight and obesity classifications. DT and SVM outperform CT in specificity by 0.8% and 6.4%, respectively, in overweight and underweight classifications.

**Sensitivity analysis.** A sensitivity analysis is conducted using the Asian BMI cutoffs for malnutrition outcomes defined in the variable description. This analysis excludes the underweight class since the standard is the same as WHO. Fig 3 illustrates that the performance rankings of the classifiers are the same for Asian and WHO standards, although the overweight and obesity cutoffs are different. Therefore, the ensemble methods examined in this study demonstrate robust predictive performance against different malnutrition benchmarks.

## Variable importance

Table 4 shows the ranking of variables in terms of their importance in predicting women's nutritional outcomes based on the specified SML algorithms. The features from each method are assigned a rank based on the variable importance summary from the respective algorithm. Any variable with a score of 1 is the most important feature in predicting an outcome. In the last column of the table, the average scores are reported for each variable. The lower the average score, the greater the importance. Considering that not all methods rank the same features, the left-hand side of the table lists more than five features for each nutritional outcome. For example, the variable importance from RT suggests that SES, partner's educational attainment, age, residence, and breastfeeding practice are the top five features in predicting BMI. However, SES, breastfeeding practice, media access, residence, and entries in the immunization roster appear to be the top five features from the RF. Each algorithm has a unique method for determining the order of importance, which is why they differ in rank. Thus, while the breastfeeding practice of women is the second most important variable in predicting BMI from RF, it is ranked tenth in LR. Nevertheless, breastfeeding practice is an essential feature for predicting BMI as it ranks third in the average value.

Based on average scores, SES, women's age, breastfeeding practice, partner's educational attainment, and media access are the top five important features in predicting BMI among

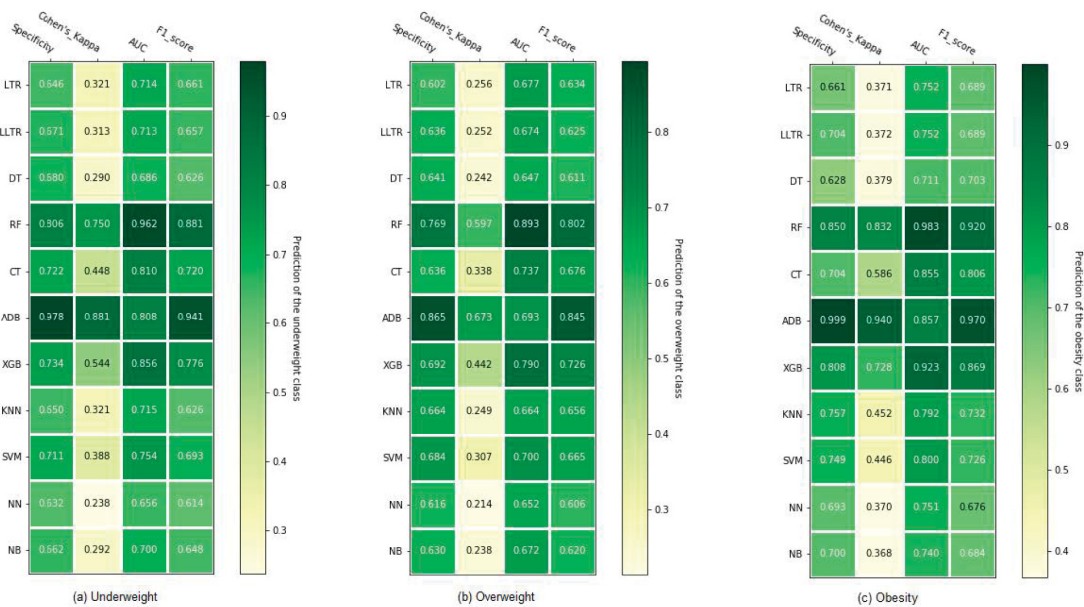

**Fig 2. Performance evaluation of different classification algorithms in predicting nutritional status using 10-fold cross-validation.** Here, LTR = Logistic regression; LLTR = LASSO logistic regression; DT = Decision tree; RF = Random forest; CT = Conditional inference tree; ADB = Adaptive boosting; XGB = eXtreme Gradient Boosting; KNN = K- nearest neighbor; SVM = Support vector machine; NN = Neural network; NB = Naïve Bayes.

women. Similarly, the prediction for the underweight class is influenced by SES, age, financial assistance to medical facilities, contraceptive methods, and breastfeeding. Additionally, SES, breastfeeding practice, partner's age, partner's educational attainment, and women's age are the five factors determining whether a woman will be overweight. Lastly, SES, media access, residence, partner's employment, and women's age are the top five variables associated with obesity. In summary, the SES, the age of the woman, and the breastfeeding practice of the woman influence the woman's nutritional status.

## Conditional inference trees

In this section, we discuss CTs generated in BMI prediction and nutritional status classification. For this, CT is chosen over CART because it provides statistical significance for both prediction and categorization. However, the visualization of the tree from the CART is presented as supplementary figures (see Supplement C in S1 File). Each CT begins with a binary split of SES. This indicates that SES is the most important feature, which is supported by Table 4.

The CT in Fig 4 shows how socioeconomic, demographic, and health variables interact to predict BMI. Nodes 14 and 15 show that the partner's educational attainment determines the BMI of women at least 26 years old from high SES. The latter node states that the average BMI is estimated to be around 26 kg/m$^2$ if their partners finished secondary or higher education and around 24 kg/m$^2$ otherwise. Breastfeeding practices also influence women's BMI, and breastfeeding women's BMI is generally lower than those of non-breastfeeding women. Additionally, their BMI are also affected by age, as nodes 7 and 8 indicate that the BMI of women from low or middle SES who are between 18 and 25 years old is lower than those over 26 years old. Furthermore, non-breastfeeding women from low SES are expected to have a lower BMI than women from middle SES.

The CT based on underweight classification is shown in Fig 5. Nodes 4 and 5 indicate that women from low SES are classified as underweight regardless of whether they use

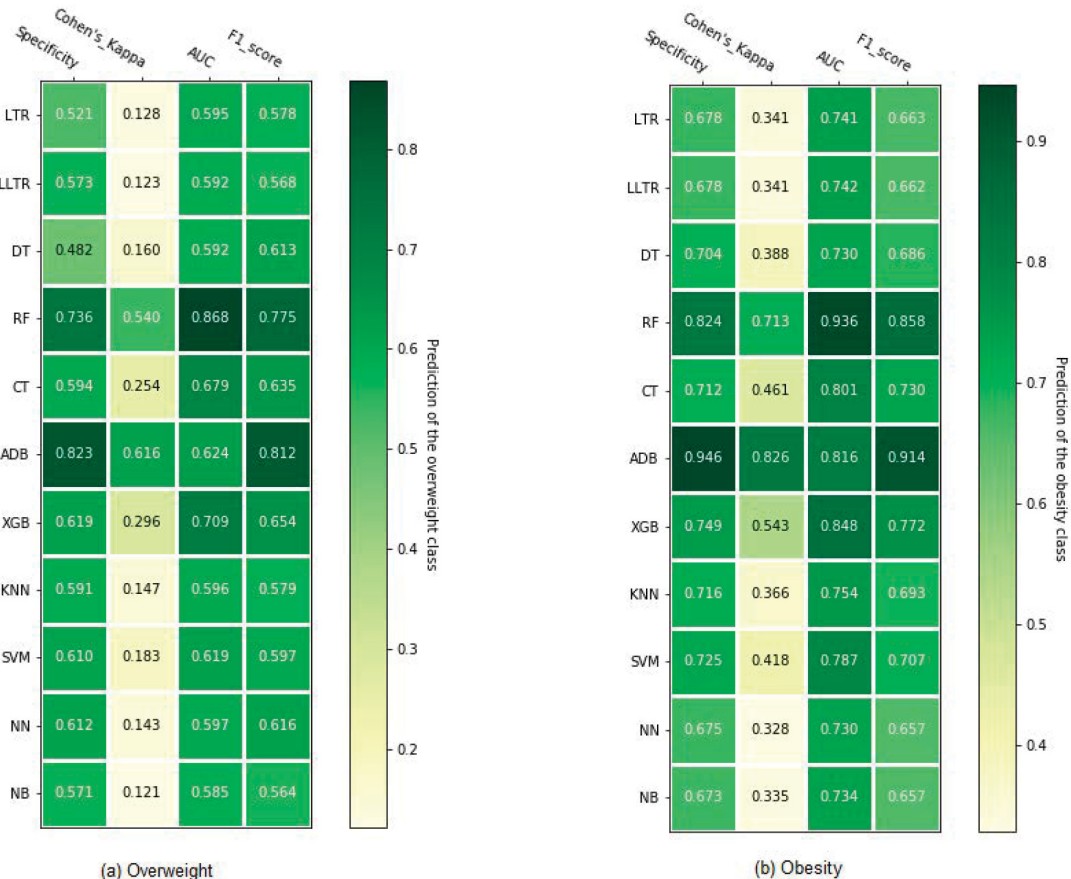

**Fig 3. Performance evaluation of different classification algorithms in predicting nutritional status based on Asian cutoff using 10-fold cross-validation.** Here, LTR = Logistic regression; LLTR = LASSO logistic regression; DT = Decision tree; RF = Random forest; CT = Conditional inference tree; ADB = Adaptive boosting; XGB = eXtreme Gradient Boosting; KNN = K-nearest neighbor; SVM = Support vector machine; NN = Neural network; NB = Naïve Bayes.

contraception. The breastfeeding practice of women from middle and high SES determines their underweight status, as seen in nodes 7, 8, 11, and 12. The latter group consists of people between the ages of 18 and 25. Breastfeeding women are categorized as underweight, whereas non-breastfeeding women are the opposite. Finally, women over 26 years from high SES are not classified as underweight, regardless of the number of children.

Fig 6 shows the CT for overweight classification. According to nodes 4, 5, 7, and 8, women from low SES are not classified as overweight regardless of whether they breastfeed or their educational level. Women from middle or high SES are classified as overweight if they live with partners at least 31 years old. Additionally, the educational attainment of their partner does not influence the classification outcome.

Fig 7 illustrates the CT for obesity classification. Nodes 4, 5, 7, and 8 show that women from middle and low SES are not categorized as obese regardless of their media access or breastfeeding practices. According to node 12, women whose partners hold either white-collar jobs or are unemployed are classified as obese, while women whose partners have blue-collar jobs are classified oppositely. The remaining two terminal nodes state that women over the age of 26 from high SES are likely to be obese.

**Table 4. Feature importance ranking from the SML algorithms.**

| Outcome/Variables | LR[C]/LTR[B] | CART[B,C] | CT[B,C] | RF[B,C] | XGB[B,C] | ADB[B] | Average |
|---|---|---|---|---|---|---|---|
| **BMI [C]** | | | | | | | |
| SES | 1 | 1 | 1 | 1 | 1 | | 1.0 |
| Age | 2 | 3 | 2 | 8 | 4 | | 3.8 |
| Breastfeeding practice | 10 | 5 | 3 | 2 | 3 | | 4.6 |
| Partner's education | 6 | 2 | 4 | 7 | 5 | | 4.8 |
| Media access | 7 | 10 | 7 | 3 | 2 | | 5.8 |
| Residence | N/A | 4 | 8 | 4 | 8 | | 6.0 |
| Employment status | 3 | 15 | 6 | 10 | 6 | | 8.0 |
| Education | 4 | 7 | 5 | 14 | 13 | | 8.6 |
| Immunization roster entry | N/A | 6 | 17 | 5 | 11 | | 9.8 |
| Total children born | 5 | 13 | 9 | N/A | 14 | | 10.3 |
| **Underweight [B]** | | | | | | | |
| SES | 1 | 1 | 1 | 1 | 1 | 1 | 1.0 |
| Age | 5 | 2 | 2 | 12 | 4 | 2 | 4.5 |
| Financial assistance | 10 | 15 | 5 | 3 | 2 | 10 | 7.5 |
| Contraceptive method | 3 | 10 | 13 | 7 | 5 | 7 | 7.5 |
| Breastfeeding practice | 9 | 5 | 6 | 16 | 8 | 3 | 7.8 |
| Education | 6 | 6 | 4 | 14 | 10 | 8 | 8.0 |
| Partner's education | 12 | 3 | 7 | 9 | 14 | 4 | 8.2 |
| Total children born | 2 | 11 | 11 | N/A | N/A | 9 | 8.3 |
| Partner's occupation | 8 | 12 | 3 | 10 | 13 | 6 | 8.7 |
| Employment status | 7 | 14 | 9 | N/A | 3 | 13 | 9.2 |
| Religion | 4 | 24 | 10 | 2 | N/A | 11 | 10.2 |
| Region | N/A | 16 | 20 | 4 | 11 | 5 | 11.2 |
| Residence | N/A | 8 | 23 | 8 | 7 | 17 | 12.6 |
| Immunization roster entry | N/A | 4 | 17 | N/A | 15 | 15 | 12.8 |
| Media access | N/A | 26 | N/A | 5 | 6 | 18 | 13.8 |
| **Overweight [B]** | | | | | | | |
| SES | 1 | 1 | 1 | 1 | 1 | 1 | 1.0 |
| Breastfeeding practice | 2 | 5 | 2 | 13 | 2 | 2 | 4.3 |
| Partner's age | 3 | 3 | 3 | 15 | 13 | 3 | 6.7 |
| Partner's education | 6 | 4 | 5 | 11 | 11 | 4 | 6.8 |
| Age | 4 | 8 | N/A | 14 | 3 | 5 | 6.8 |
| Employment status | 5 | 16 | 6 | 4 | 5 | 8 | 7.3 |
| Media access | 11 | 2 | N/A | 8 | 6 | 11 | 7.6 |
| Education | 7 | 7 | 4 | 12 | 10 | 6 | 7.7 |
| Distance to health facility | N/A | 11 | N/A | 3 | 4 | 19 | 9.3 |
| Region | N/A | N/A | 8 | 2 | 12 | 18 | 10.0 |
| Contraceptive method | N/A | N/A | N/A | 5 | 14 | 21 | 13.3 |
| **Obesity [B]** | | | | | | | |
| SES | 1 | 1 | 1 | 1 | 1 | 1 | 1.0 |
| Media access | 2 | 2 | 3 | 2 | 2 | 3 | 2.3 |
| Residence | 6 | 3 | 4 | 4 | 4 | 6 | 4.5 |
| Partner's employment | 4 | 5 | 9 | 3 | 3 | 5 | 4.8 |
| Age | 3 | 7 | 2 | 10 | 8 | 2 | 5.3 |
| Employment status | 7 | N/A | 7 | 5 | 5 | 12 | 7.2 |
| Partner's age | 5 | 8 | 8 | 11 | 14 | 4 | 8.3 |

*(Continued)*

**Table 4.** (Continued)

| Outcome/Variables | LR^C/LTR^B | CART^B,C | CT^B,C | RF^B,C | XGB^B,C | ADB^B | Average |
|---|---|---|---|---|---|---|---|
| Region | 8 | N/A | 6 | 12 | 6 | 10 | 8.4 |
| Partner's education | 11 | 4 | 5 | 14 | 13 | 7 | 9.0 |
| Education | 12 | 6 | 12 | 15 | 11 | 9 | 10.8 |
| Breastfeeding practice | 16 | N/A | 14 | N/A | N/A | 8 | 12.7 |

* B: Binary outcome, C: Continuous outcome, LR: Linear regression, LTR: Logistic regression, CART: Classification and regression tree, CT: conditional tree, RF: Random Forest, XGB: eXtreme Gradient Boosting, ADB: Adaptive boosting. N/A means the rank is not available from the SML algorithm.

## Discussion

Our initial descriptive statistics suggest that the BMI values of sampled women range between 22.0 and 25.4 kg/m² , and the average is 23.5 kg/m² . Our bivariate analyses also indicate that BMI is significantly and positively associated with age and educational attainment (of both women and their partners), which is in line with recent studies from Bangladesh [80–82]. We find that breastfeeding women have a lower BMI than their non-breastfeeding counterparts, which (i.e., the protective effect of breastfeeding against high BMI) is also in line with studies in Bangladesh and elsewhere [83–86].

Currently, about 27% of the sampled women appear to be overweight and about 7% obese, compared with only 11% being underweight. This nutrition transition among women of child-bearing age in Bangladesh from undernutrition to overnutrition was identified in the mid-2010s [87, 88], and now it is established that the proportion of women being overweight/obese in Bangladesh surpassed the proportion of women underweight in around 2010 [89]. Both from nationally representative BDHS data and findings from other recent studies support this transition [90–92]. From our bivariate analyses, the estimated prevalence of overweight is nearly double among women from high SES, women who did not breastfeed their children, and older (i.e., age 26 years or over) compared to women from low SES, women who

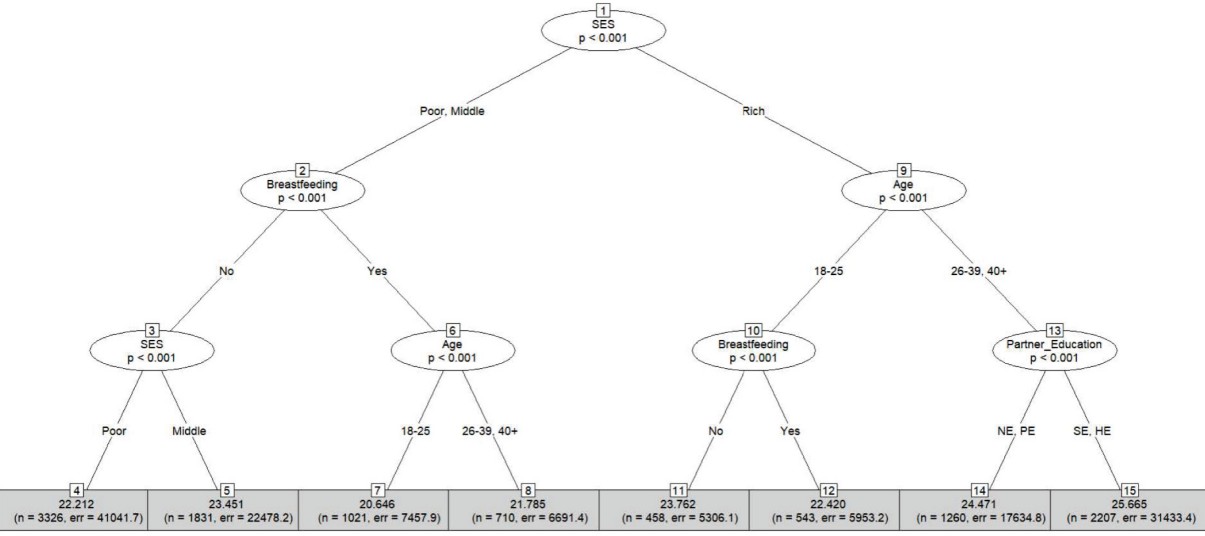

**Fig 4. Conditional inference tree for BMI prediction using 10-fold cross-validation.** Each terminal node of this algorithm provides the average predicted value, the number of observations used, and the sum of squared error. NE, PE, SE, and HE represent no education, primary education, secondary education, and higher degree (such as college), respectively.

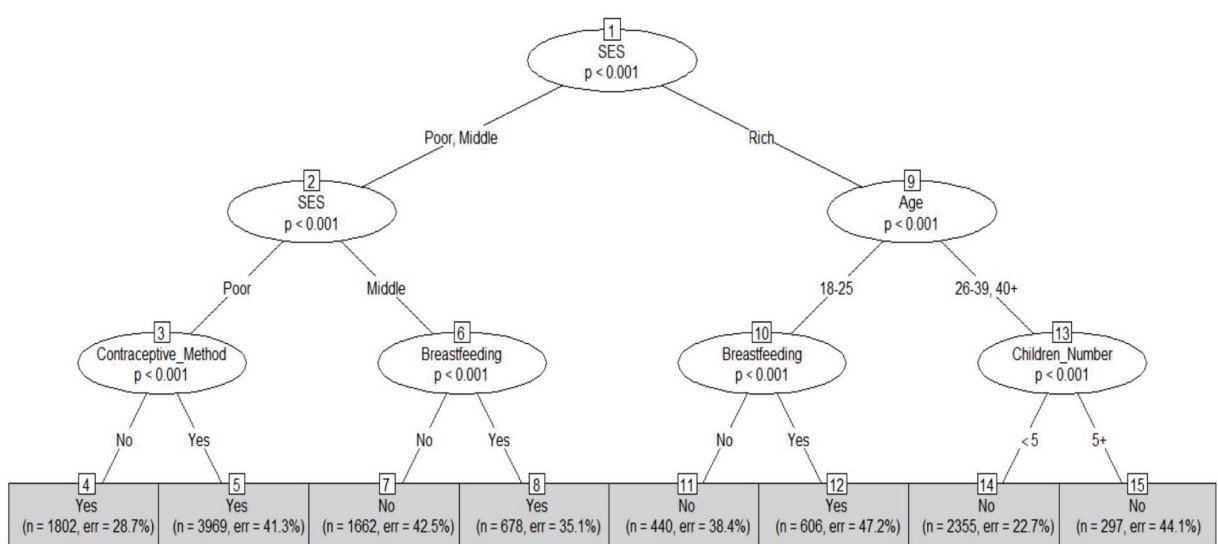

**Fig 5. Conditional inference decision tree for underweight using 10-fold cross-validation.** Each terminal node of this algorithm provides the underweight status (Yes or No), the number of observations used, and the classification error.

breastfeed their children, and age 25 years or less, respectively. In addition, the prevalence of obesity is nearly four times higher among women from high SES compared to women from low SES. Women with higher education and older (i.e., aged 26 years or above) have a three times higher prevalence of obesity compared to those without education and aged 25 years or less, respectively. The percentage of obesity among women who do not breastfeed their children is twice that of those who do. Recent studies in Bangladesh also confirm that age, SES, and educational attainment are statistically significant predictors of overweight and obesity among women of childbearing age [80, 82, 93–95].

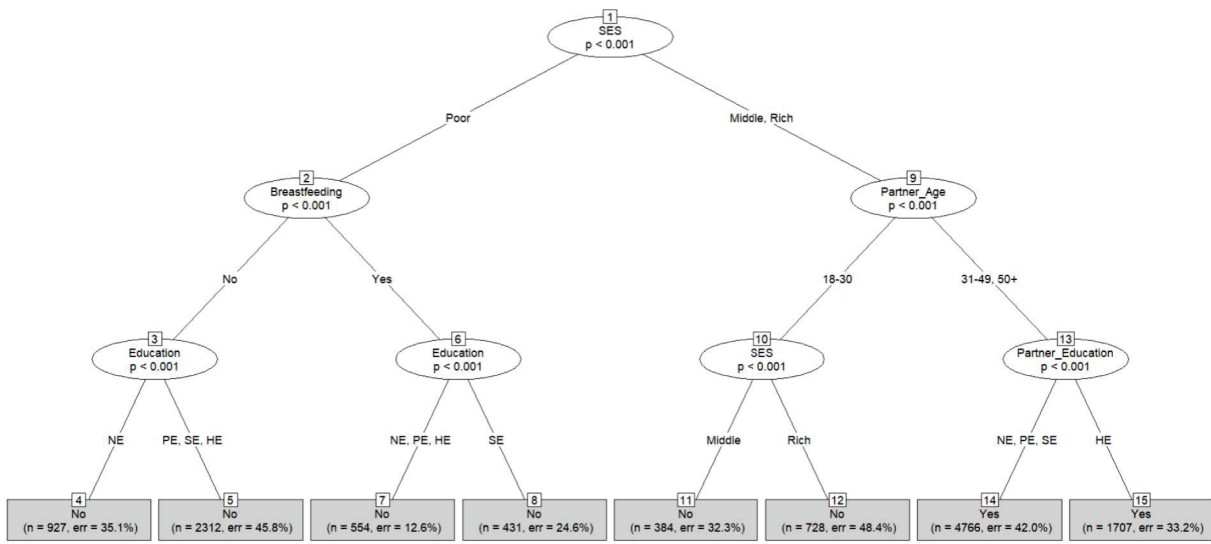

**Fig 6. Conditional inference decision tree for overweight using 10-fold cross-validation.** The categories of the education variable are the same as those of partner's education. Each terminal node of this algorithm provides the overweight status (Yes or No), the number of observations used, and the classification error.

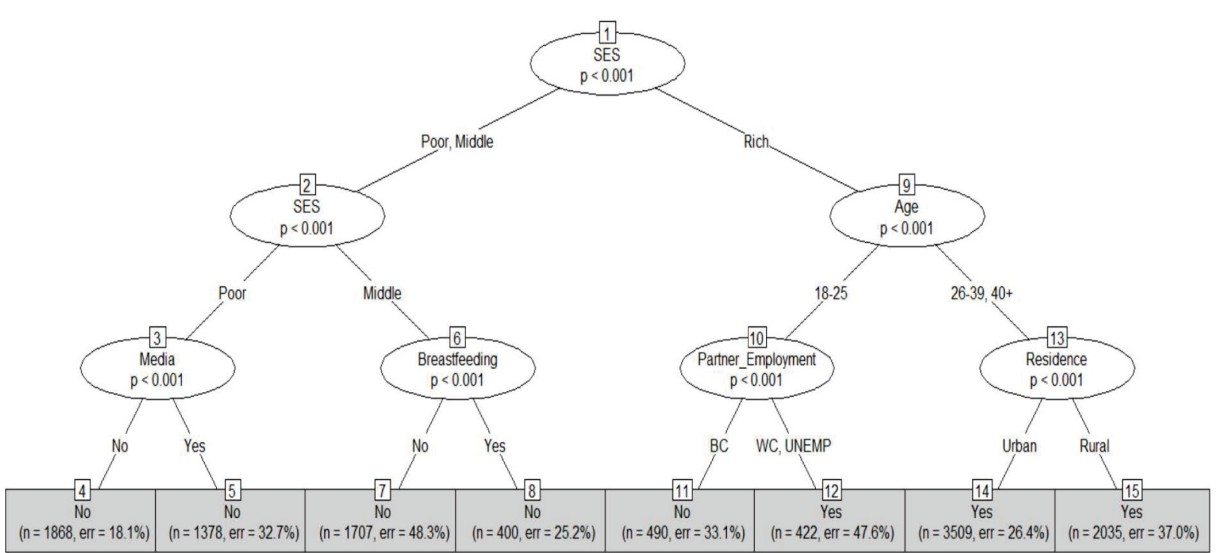

**Fig 7. Conditional inference decision tree for obesity using 10-fold cross-validation.** On the partner's employment variable, BC, WC, and UNEMP respectively represent blue-collar job, white-collar-job, and unemployed. Each terminal node of this algorithm provides the obesity status (Yes or No), the number of observations used, and the classification error.

In our BMI prediction, the top two algorithms are SVM and KNN in terms of $R^2$, RMSE, and MAE. Specifically, the $R^2$ values of SVM and KNN exceed that of LR by 8.6% and 5.5%, respectively. In underweight, overweight, and obesity classifications, the combined predictor algorithms consistently yield top specificity, Cohen's kappa, F1-score, and AUC, and their performance is robust across different outcome standards. We compare our results with those from previous studies that applied SML to classify different health outcomes (see SD-1 Table in S1 File). Our classification models for underweight and overweight outperform those of other studies in terms of accuracy values (94% and 84%). Additionally, the AUCs from all three classifications (96.2%, 89.3%, and 98.3%) of this paper based on RF exceed those of other studies reported in SD-1 Table in S1 File. All these results verify the strength of our models. Figueroa and Flores achieved a slightly better accuracy (97.36%) in obesity classification based on electronic medical records using SVM [61]. Although their study used a much larger sample size and a different cohort, the accuracy of the classification model is very comparable, demonstrating the generalization of the training model and the representative power of the samples.

Feature importance ranking based on selected algorithms suggests that SES, women's age, and breastfeeding status are strongly associated with nutritional outcomes. Specifically, SES is the most important as it ranks first across all methods. The CT for BMI prediction using 10-fold cross-validation demonstrates that the average predicted BMI value among women of childbearing age in Bangladesh increases significantly (p<0.001) with SES (i.e., low vs. middle and low+middle vs. high), age (i.e., 18–25 vs. 25+ years), and breastfeeding practices (i.e., women who breastfeed their babies vs. who do not). Also, among the women from high SES, their partner's educational attainment (i.e., no or primary education vs. secondary education or higher) significantly increases the predicted BMI value (p<0.001). CTs in our study have shown that SES strongly influences nutritional outcomes among women of childbearing age in Bangladesh. Additionally, both the age of women and breastfeeding practice appeared consistently on the trees, indicating that these variables are also important in predicting nutrition outcomes. Furthermore, the partner's educational attainment influences BMI prediction and

overweight classification. Women's obesity risk is associated with their partner's employment and media access. These findings correspond with the results from previous studies in Bangladesh and elsewhere [15, 82, 96].

## Strength and limitations

This study makes several notable contributions to the existing literature on BMI and malnutrition outcomes in women of childbearing age. First, unlike most previous studies that utilized only standard LR or ordinal categorical response models to estimate the associations of socioeconomic determinants with malnutrition outcomes in women, this study implements a variety of SML methods to predict the BMI value as well as nutritional status, namely underweight, overweight, and obesity. To our knowledge, as of now, only two studies attempted to predict the risks of malnutrition outcomes in adults in Bangladesh [64, 65]. Nevertheless, neither predicted the BMI outcome nor defined how features interact and lead to prediction, aside from evaluating the performance of models. Additionally, the study by Ferdowsy and colleagues [64] had limited generalizability due to its small sample size ($n = 1,100$), and the study by Islam and colleagues [65] used data from an older round (i.e., 2014) of BDHS [64].,. We overcome these issues by utilizing data from the most recent wave of BDHS (i.e., 2017–18), a nationally representative survey with about 15,000 observations, and by exploring how hidden interactions among features influence the prediction of key outcomes using tree-based approaches. Hence our findings are generalizable at the national level and provide updated information for policy and programmatic considerations.

Second, the feature selection used in our study eliminates unimportant regressors, shortens the computational time and cost, and strengthens the accuracy of the predictive model. A number of studies identified the important risk factors for underweight and overweight/obesity status in women from traditional logistic or multinomial logistic regression [65, 97, 98]. They selected those features that appeared significant in the corresponding regression models. Unlike them, this study pins important features and ranks them using SML algorithms. SML feature selection algorithms are deemed robust for choosing features in the model [99, 100].

Third, due to a small number of observations under the obesity category, all previous studies using BDHS data modeled overweight and obesity as a single indicator variable [15, 65, 82, 101–103]. This study is the first that attempts to model them separately to unmask the risk associated with obesity and distinguish it from that of being overweight. We overcome the issue of an unbalanced distribution of the obesity category (i.e., the percentage of obese women in this study sample is only about 7%, which can potentially lead to biased and unreliable results while using SML) by generating a synthetic balanced sample proposed by Menardi and colleagues [75, 104].

Lastly, this study favors CT over CART since the former has some relative advantages. CART is the typical approach to generating a tree that maps the features to the predicted value. CT builds the tree using a conditional distribution, which captures the relationship between features and outcomes. Unlike CART, this approach creates a tree by conducting the variable independence test. By examining the conditional distribution, this test identifies whether a particular feature significantly predicts both continuous and categorical outcomes.

In terms of limitations, the scope of our analyses relies on variables included in the BDHS cross-sectional dataset. The performance of our predictions and feature importance rankings can potentially be enhanced by adding relevant covariates (e.g., dietary practices by women of childbearing age), which are currently not in the BDHS dataset. Also, the wealth index used in this analysis as a socioeconomic feature can potentially cause misclassification bias in the measurement of SES. Several studies have discussed the limitations of the wealth index in the DHS

and its potential for misclassification bias [105–107]. The wealth index is constructed based on household assets and amenities [108]. While it is a well-accepted and useful proxy for wealth and economic status, it may not capture other important dimensions of socioeconomic status, such as income, education, and occupation. Finally, this paper focuses on Bangladeshi child-bearing women aged from 18 to 49 years. For this reason, we excluded 5,365 sampled respondents from our empirical analyses who were either under the age of 18, or with missing data, or considered irrelevant observations/outliers based on Cook's distance to prevent the loss of prediction strength captured by the $R^2$ [109]. All observations with Cook's distance greater than 4/n (n is the number of rows) after running a linear regression are taken out from the analysis as they are considered outliers [110]. The regression result is improved after eliminating them as the $R^2$ enhances to 22% from 17.5%, which was the $R^2$ value with outliers. Exclusion of the sampled respondents, particularly for missing values, can potentially cause selection bias [111].

## Conclusion

To the best of our knowledge, this is the first study that predicts BMI and one of the pioneer studies to classify all three malnutrition outcomes for women of childbearing age in Bangladesh, let alone in any lower-middle income country, using SML techniques. Moreover, our paper is the first to identify and rank a set of socioeconomic, health, and demographic features that are critical in predicting nutritional measures using several SML feature selection algorithms. The estimators from this study predict the outcomes of interest most accurately and efficiently compared to other existing studies in the relevant literature. Therefore, study findings can aid policymakers in designing policy and programmatic approaches to address the double burden of malnutrition among Bangladeshi women, thereby reducing the country's economic burden. Moreover, our findings would be valuable for identifying women at risk for any form of malnutrition to design and implement tailored interventions in the future.

The policy documents on women's health and nutrition in Bangladesh largely focus on the undernutrition aspect of malnutrition [89]. The National Nutrition Policy 2015 identified 'reducing maternal overweight' to be critical for achieving optimal nutrition, and the National Strategy for Maternal Health 2019–2030 acknowledged the increasing trends in overweight and obesity, but neither provided any specific programmatic directions for addressing overweight and obesity among women of childbearing age in the country [112]. The Second National Plan of Action for Nutrition (NPAN-2) 2016–2025 has outlined some activities for prevention and control of obesity and non-communicable diseases, without specifying the target groups for each of these activities [89, 113]. Based on the findings of this study and the 'best practices' for reducing overweight and obesity in other countries from South and Southeast Asia, we conclude with the following policy and programmatic implications:

1. Scaling up breastfeeding: the benefits of breastfeeding for a healthier weight and postpartum weight loss have been well-documented [86, 114, 115]. However, breastfeeding practice is still not optimum in Bangladesh, and women face a range of barriers at the individual, societal, and system levels that include adverse effects of breastfeeding on maternal health, nutrition, and physical appearance [116]. Proven interventions for scaling up breastfeeding in Bangladesh and other Asian countries, which combined intensified interpersonal counseling, mass media campaigns, and community mobilization interventions [117], need to be implemented not only for ensuring nutrition for children but also as an effective approach for healthy BMI for women of childbearing age.

2. Designing culturally adapted community-based physical activity interventions for women of childbearing age: existing evidence indicates that culturally adapted physical activity interventions tended to be more accepted by the target community, particularly for women, and be effective in reducing weight among South Asian adults [118, 119]. Using participatory approaches to identify relevant cultural attitudes and norms to inform the design of community-based physical activity interventions would improve physical activity uptake and/or adherence for older, better-off women. The government's National Nutrition Services can consider building partnership with the private/non-governmental sector for urban areas and recruiting community health workers in rural areas to promote and sustain physical activities among women of childbearing age since studies found that trained local facilitators help overcome the barriers to physical intervention uptake in among Bangladeshi and other South Asian descendants [118, 120, 121].

3. Implementing behavior change communications for diet and lifestyle modifications for urban, high SES adults: studies indicate that routine consultations and lifestyle advice from physicians are ineffective in modifications in diet and lifestyle that result in viable weight loss [122]. However, a study on individuals of South Asian descent living in the United Kingdom found that family-based lifestyle intervention consisting of about 15 visits from a dietitian over three years resulted in significantly greater weight loss than did annual contact and simple lifestyle advice from a dietitian [123]. In our analyses, both RT and RF highlighted high SES and living in urban areas to be among the top five features in predicting BMI. For this reason, family-based lifestyle intervention involving dieticians can be explored for urban, affluent women.

4. Implementing recent legislations on nutrition: the Government of Bangladesh has published several regulations to promote healthy diets (viz., Food Safety [Labelling] Regulations 2017, Breastmilk Substitute [BMS] Rules 2017, Limiting Trans Fatty Acids in Foodstuffs Regulations 2021) in the recent years. These regulations need to be implemented to control the marketing of junk/packaged foods and sugary beverages and their consumption.

5. Setting up an effective nutrition surveillance system: both the National Nutrition Policy 2015 and NPAN-2 acknowledged the need for setting up a nutrition surveillance system to track nutritional status and identify locations for targeted nutrition programs [113, 124]. The government and other relevant stakeholders in the country need to work together and set up a community-based surveillance system with adequate coverage without delay. We agree with Khan and colleagues that tracking malnutrition is particularly crucial for adolescent girls, who remain vulnerable to both forms of malnutrition [89].

## Supporting information

**S1 File.**
(DOCX)

## Acknowledgments

We acknowledge the institutions and individuals who made the 2017–18 Bangladesh Demographic and Health Survey possible. The 2017–18 BDHS was implemented under the authority of the National Institute of Population Research and Training (NIPORT) of the Government of the People's Republic of Bangladesh with financial assistance from USAID/Bangladesh. The 2017–18 BDHS data were collected and processed by Mitra & Associates. Lastly, the authors

would like to thank the participants of the Bangladesh Demographic and Health Surveys who participated in this study.

## Author Contributions

**Conceptualization:** Md. Mohsan Khudri.

**Data curation:** Md. Mohsan Khudri, Kang Keun Rhee.

**Formal analysis:** Md. Mohsan Khudri, Kang Keun Rhee, Mohammad Shabbir Hasan.

**Methodology:** Md. Mohsan Khudri, Kang Keun Rhee, Karar Zunaid Ahsan.

**Supervision:** Karar Zunaid Ahsan.

**Visualization:** Mohammad Shabbir Hasan.

**Writing – original draft:** Md. Mohsan Khudri, Kang Keun Rhee, Karar Zunaid Ahsan.

**Writing – review & editing:** Md. Mohsan Khudri, Mohammad Shabbir Hasan, Karar Zunaid Ahsan.

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
