## [Decision Letter · Decision Letter 0]

27 Mar 2023

PONE-D-22-29868Predicting nutritional status for women of childbearing age from their economic, health, and demographic features: A supervised machine learning approachPLOS ONE

Dear Dr. Ahsan,

Thank you for submitting your manuscript to PLOS ONE. After careful consideration, we feel that it has merit but does not fully meet PLOS ONE’s publication criteria as it currently stands. Therefore, we invite you to submit a revised version of the manuscript that addresses the points raised during the review process.

We look forward to receiving your revised manuscript.

Kind regards,

Tauseef Ahmad

Academic Editor

PLOS ONE

Journal Requirements:

Additional Editor Comments:

We have now received two reviewers’ reports for your manuscript. The reviewers praise aspects of your work, however they also point out some issues. The reviewers’ points are clearly laid out in their reviews which are appended below.

Reviewers' comments:

Reviewer's Responses to Questions

**Comments to the Author**

1. Is the manuscript technically sound, and do the data support the conclusions?

Reviewer #1: Yes

Reviewer #2: Yes

2. Has the statistical analysis been performed appropriately and rigorously? 

Reviewer #1: Yes

Reviewer #2: I Don't Know

3. Have the authors made all data underlying the findings in their manuscript fully available?

Reviewer #1: No

Reviewer #2: Yes

4. Is the manuscript presented in an intelligible fashion and written in standard English?

Reviewer #1: Yes

Reviewer #2: Yes

5. Review Comments to the Author

Reviewer #1: The value of the wealth index relies on the assets mentioned above owned by households and housing characteristics. However, this study does not suggest asset index as a measure of current welfare or poverty in this paper. Instead, we use it as a determinant of malnutrition among women, which depends on the long run as well as current SES. – this information might have misclassification bias, which was not mentiioned as a limitation.

5,365 women are removed out of 20,127 from the analysis – there is no explanation for the reason to exclude 5,365 women – this might have selection bias, which was not mentiioned as a limitation.

This study evaluates the relationship between categorical features and nutritional outcomes using the Pearson statistic corrected for the complex survey design with the second-order correction of Rao and Scott – This sentence had been presented in the results section and it should be in the methods section.

There is a statistically significant association between each feature and underweight status, except for abstinence, family planning, and the number of family members. – this sentence is not very clear – it needs to be reviewed.

A mere 5% of the sampled women from high SES are underweight, compared to nearly threefold among those from low socioeconomic backgrounds. – in the result section, no need to use adjective.

Among breastfeeding women, 16% are underweight compared to 9% among non-breastfeeding women. – no statistic had been presented. And the following comparisons in this section did not present statistical analysis, as well.

Reviewer #2: This paper is about Body Max Index and various nutritious rank a set of socioeconomic, health, and demographic features that are critical in predicting nutritional measures several SML feature selection algorithms. The authors have used machine Learning tools which I am not good. Apparently, it looks ok but someone who is expert in Machine learning and artificial intelligence may comment better.

6. PLOS authors have the option to publish the peer review history of their article (what does this mean?). If published, this will include your full peer review and any attached files.

Reviewer #1: **Yes: **Airton Tetelbom Stein

Reviewer #2: No

---

## [Author Response · Author response to Decision Letter 0]

19 Apr 2023

Comments from Reviewer 1

1. The value of the wealth index relies on the assets mentioned above owned by households and housing characteristics. However, this study does not suggest asset index as a measure of current welfare or poverty in this paper. Instead, we use it as a determinant of malnutrition among women, which depends on the long run as well as current SES. – this information might have misclassification bias, which was not mentiioned as a limitation.

Authors’ Response: We added this as a limitation on p.37, lines 735–741.

2. 5,365 women are removed out of 20,127 from the analysis – there is no explanation for the reason to exclude 5,365 women – this might have selection bias, which was not mentioned as a limitation.

Authors’ Response: We explained the exclusions on p.17, lines 370–373. We also added this as a limitation on p.37, lines 741–749. 

3. This study evaluates the relationship between categorical features and nutritional outcomes using the Pearson statistic corrected for the complex survey design with the second-order correction of Rao and Scott – This sentence had been presented in the results section and it should be in the methods section.

Authors’ Response: We moved the sentence to the methods section (p.17, lines 373–376). 

4. There is a statistically significant association between each feature and underweight status, except for abstinence, family planning, and the number of family members. – this sentence is not very clear – it needs to be reviewed.

Authors’ Response: We thank the reviewer for highlighting this. We revised the whole section (pp.20–21, lines 443–467) to make this clearer. 

5. A mere 5% of the sampled women from high SES are underweight, compared to nearly threefold among those from low socioeconomic backgrounds. – in the result section, no need to use adjective.

Authors’ Response: We dropped the adjective and revised the section (pp.20–21, lines 443–467) for clarification. 

6. Among breastfeeding women, 16% are underweight compared to 9% among non-breastfeeding women. – no statistic had been presented. And the following comparisons in this section did not present statistical analysis, as well.

Authors’ Response: We revised the section (pp.20–21, lines 443–467) for clarification. 

Comment from Reviewer 2

1. This paper is about Body Max Index and various nutritious rank a set of socioeconomic, health, and demographic features that are critical in predicting nutritional measures several SML feature selection algorithms. The authors have used machine Learning tools which I am not good. Apparently, it looks ok but someone who is expert in Machine learning and artificial intelligence may comment better.

Authors’ Response: We thank the reviewer for the careful and insightful review of our manuscript. We used a variety of machine learning algorithms that were used in other health outcome-related studies to derive the results of our study. For instance, Daoud et al. (2019) implemented neural network, linear regression, LASSO regression, random forest, and CART (classification and regression trees) to predict women’s height across 66 nations. DeGregory et al. (2018) attempted logistic regression, decision trees, and neural network to classify the following health outcomes: high blood pressure and body fat. Yadaw et al. (2020) showed that eXtreme Gradient Boosting produced the highest area under the repository curve (AUC) to predict deaths from COVID-19 in the U.S. Rahman et al. (2021) adopted logistic regression, random forest, and support vector machine to predict childhood stunting prevalence in Bangladesh and stated that random forest yielded the best classification results. Lastly, Peterson et al. (2016) used conditional tree analysis to study the association between muscle strength and cardiometabolic risk.

In addition to the algorithms adopted in the aforementioned studies, we also attempted adaptive boosting, k-nearest neighbor, and Naïve Bayes to make our analysis more comprehensive and robust.

References

1. Daoud A, Kim R, Subramanian SV. Predicting women's height from their socioeconomic status: A machine learning approach. Social Science & Medicine. 2019 Oct 1;238:112486.

2. DeGregory KW, Kuiper P, DeSilvio T, Pleuss JD, Miller R, Roginski JW, Fisher CB, Harness D, Viswanath S, Heymsfield SB, Dungan I. A review of machine learning in obesity. Obesity reviews. 2018 May;19(5):668-85.

3. Yadaw AS, Li YC, Bose S, Iyengar R, Bunyavanich S, Pandey G. Clinical features of COVID-19 mortality: development and validation of a clinical prediction model. The Lancet Digital Health. 2020 Oct 1;2(10):e516-25.

4. Rahman SJ, Ahmed NF, Abedin MM, Ahammed B, Ali M, Rahman MJ, Maniruzzaman M. Investigate the risk factors of stunting, wasting, and underweight among under-five Bangladeshi children and its prediction based on machine learning approach. Plos one. 2021 Jun 17;16(6):e0253172.

5. Peterson MD, Zhang P, Saltarelli WA, Visich PS, Gordon PM. Low muscle strength thresholds for the detection of cardiometabolic risk in adolescents. American journal of preventive medicine. 2016 May 1;50(5):593-9.

---

## [Editor Report · Decision Letter 1]

2 May 2023

Predicting nutritional status for women of childbearing age from their economic, health, and demographic features: A supervised machine learning approach

PONE-D-22-29868R1

Dear Dr. Ahsan,

We’re pleased to inform you that your manuscript has been judged scientifically suitable for publication and will be formally accepted for publication once it meets all outstanding technical requirements.

Kind regards,

Tauseef Ahmad

Academic Editor

PLOS ONE

Additional Editor Comments (optional):

My appreciation goes out to the authors for their excellent revisions, as well as the English editing.
---

## [Editor Report · Acceptance letter]

5 May 2023

PONE-D-22-29868R1 

Predicting nutritional status for women of childbearing age from their economic, health, and demographic features: A supervised machine learning approach 

Dear Dr. Ahsan:

I'm pleased to inform you that your manuscript has been deemed suitable for publication in PLOS ONE. Congratulations! Your manuscript is now with our production department. 

Kind regards, 

on behalf of

Dr. Tauseef Ahmad 

Academic Editor

PLOS ONE